# Effect of Metabolic Adaptation by Voluntary Running Wheel Activity and Aldosterone Inhibition on Renal Function in Female Spontaneously Hypertensive Rats

**DOI:** 10.3390/cells11243954

**Published:** 2022-12-07

**Authors:** Felix Atmanspacher, Rolf Schreckenberg, Annemarie Wolf, Ivica Grgic, Klaus-Dieter Schlüter

**Affiliations:** 1Physiologisches Institut, Justus-Liebig-Universität Gießen, 35392 Gießen, Germany; 2Klinik für Nephrologie und Transplantationsmedizin, Philipps Universität Marburg, 35043 Marburg, Germany

**Keywords:** PCSK9, arginase-2, microalbuminuria, Kim-1, blood urea nitrogen, voluntary wheel running, spironolactone

## Abstract

Metabolic effects of physical activity may be reno-protective in the context of hypertension, although exercise stresses kidneys. Aldosterone participates in renal disease in hypertension, but exercise affects the plasma concentration of aldosterone. This study was designed to evaluate whether physical activity and pharmacological treatment by aldosterone have additive effects on renal protection in hypertensive rats. Female spontaneously hypertensive rats (SHR) or normotensive Wistar rats performed voluntary running wheel activity alone or in combination with aldosterone blockade (spironolactone). The following groups were studied: young and pre-hypertensive SHR (n = 5 sedentary; n = 10 running wheels, mean body weight 129 g), 10-month-old Wistar rats (n = 6 sedentary; n = 6 running wheels, mean body weight 263 g), 10-month-old SHRs (n = 18 sedentary, mean body weight 224 g; n = 6 running wheels, mean body weight 272 g; n = 6 aldosterone, mean body weight 219 g; n = 6 aldosterone and running wheels, mean body weight 265 g). Another group of SHRs had free access to running wheels for 6 months and kept sedentary for the last 3 months (n = 6, mean body weight 240 g). Aldosterone was given for the last 4 months. SHRs from the running groups had free access to running wheels beginning at the age of 6 weeks. Renal function was analyzed by microalbuminuria (Alb/Cre), urinary secretion of kidney injury molecule-1 (uKim-1), and plasma blood urea nitrogen (BUN) concentration. Molecular adaptation of the kidney to hypertension and its modification by spironolactone and/or exercise were analyzed by real-time PCR, immunoblots, and histology. After six months of hypertension, rats had increased Alb/Cre and BUN but normal uKim-1. Voluntary free running activity normalized BUN but not Alb/Cre, whereas spironolactone reduced Alb/Cre but not BUN. Exercise constitutively increased renal expression of proprotein convertase subtilisin/kexin type 9 (PCSK9; mRNA and protein) and arginase-2 (mRNA). Spironolactone reduced these effects. uKim-1 increased in rats performing voluntary running wheel activity exercise irrespectively of blood pressure and aldosterone blockade. We observed independent but no additive effects of aldosterone blockade and physical activity on renal function and on molecules potentially affecting renal lipid metabolism.

## 1. Introduction

A high level of physical activity is part of a healthy lifestyle [1]. However, exercise itself can stress the kidney, leading to molecular adaptations of the renin-angiotensin-aldosterone system (RAAS) [2,3]. Another example by which exercise affects kidney function is the decrease in plasma carnitine levels that leads to an up-regulation of organic cation transporter (OCTN2) on mRNA and protein levels, indicating acute changes in transcriptional adaptation to exercise in the kidneys of mice [4]. In rats performing moderate activity for five weeks, plasma sodium, potassium, and osmolality increased in comparison to sedentary rats suggesting alterations in ion homeostasis that is regulated by the kidney [5]. In another study, rats performing swimming developed a decrease in creatinine clearance and an increase in urinary sodium secretion together with changes in angiotensin receptor type 1 (AT_1_R) expression in the kidney [6]. In an analysis of 22 participants of the Boston Marathon in 2015, it was found that acute kidney injury occurred during the competition with a subsequent indication of albuminuria but was also repaired on the following day [7]. Proteinuria in response to exercise was also observed in children [8] and patients with type-1 diabetes [9]. Acute exercise increases plasma potassium concentration in hypertensive patients with normal renal function [10]. Specifically, in young patients with mild hypertension, exercise was again associated with albuminuria suggesting early glomerular leakage [11]. These examples show that exercise can affect renal function in mice, rats, and humans with healthy or disease status. On the other hand, exercise can reduce renal fibrosis and improve renal nitric oxide metabolism [12,13]. Previously, a study from our own group provided evidence that a downregulation of the parathyroid hormone-related protein-receptor type 1 contributes to the molecular adaptations of the kidney associated with voluntary running wheel activity in spontaneously hypertensive rats [14]. Collectively, stress during the acute phase of exercise might translate into adaptive mechanisms when performed for a longer time. The question that arose from these findings summarized above is how these reactions of the kidney to high physical activity interact with the additional burden of hypertension.

Spontaneous hypertensive rats (SHR) are a preferred animal model for investigating the effects of physical activity in the concomitant presence of hypertension. Data obtained from this model suggest that exercise can protect the kidney structure [15], can reduce albuminuria [16] as well as the expression of components of the renin-angiotensin system (RAS) [17], and is capable to increase the expressions of UCP-2 and SOD-2, though both of them are considered as potential mechanisms reducing oxidative stress [18]. Moreover, voluntary running wheel activity affects subcutaneous white adipose tissue in these rats indicating tissue-specific metabolic effects [19]. However, limitations of these studies are the short training periods in most of the underlying protocols. To study the effect of an active life style on kidney function in hypertensive rats it should be kept in mind that end-organ damage occurs at later time points. Furthermore, in most cases protective effects were associated with a reduction in blood pressure (i.e., ref. [15]). However, in voluntary running wheel activity over a period of several months, an experimental protocol that mimics a life-long active life style during aging, displayed no effect on blood pressure in SHRs but significant effects on kidney [14,20,21,22]. Moreover, swimming affects the RAS differentially in young and adult rats [23]. Collectively, the question how the already stressed kidney of hypertensives responses to prolonged high physical activity is not yet clarified and therefore, it is still not possible to predict the effect of high physical activity on renal function in hypertensives with respect to the interaction between blood pressure, aging, and exercise.

Aldosterone might play a pivotal role in the pathophysiology of the kidney in hypertensives. High aldosterone levels are strongly associated with microalbuminuria, a risk factor in the context of kidney disease [24,25,26]. Mechanistically aldosterone may affect the glycocalyx via modification of matrix metalloprotease (MMP)-2 and MMP-9 activity [27]. There is some evidence that exercise may lower plasma concentration of aldosterone, an effect of exercise that is of specific interest in hypertensives with an activated RAAS [28]. Interestingly, microalbuminuria develops under high aldosterone and exercise but exercise-induced microalbuminuria may be less relevant for disease progression [29]. These interesting interactions prompted us to investigate the combined effect of aldosterone blockade and free running wheel activity in SHRs.

In the current study, we analyzed kidney structure and function in SHRs that performed voluntary running wheel activity for 10 months. As we found functional alterations such as urinary KIM-1 secretion, increased urinary albumin/creatinine, and increased plasma blood urea nitrogen (BUN) in the absence of severe morphological alterations, we first screened molecular markers that display an early adaptation to voluntary running wheel activity in young (still pre-hypertensive) SHRs. Next, we analyzed which of the markers remained elevated after months of voluntary running wheel activity. As exercise is known to reduce the level of aldosterone [28], we tested the hypothesis that the level of physical activity directly affects the molecular adaptations of the kidney to hypertension and that these effects are independent of aldosterone. Overall, we identified proprotein convertase subtilisin/Kexon type 9 (PCSK9), a molecule involved in fatty acid metabolism, and arginase-2 as molecules that were induced early by physical activity in hypertensive rats and remained elevated when constantly proceeding with high physical activity.

## 2. Materials and Methods

The investigations are in agreement with the “Guide for the Care and Use of Laboratory Animals” purchased by the U.S. National Institute of Health (NIH Publication No. 85-23, revised 1996). The study was approved by the local authorities (RP Gießen; V 54–19 c 20 15 h 01 GI 20/1 Nr. 76/2014 and GI 20/1 Nr. 77/2014).

### 2.1. Animal Model

The current project aimed at studying the effects of long-term high physical activity (10 months of voluntary running wheel activity) on renal function and adaptation in female spontaneously hypertensive rats (n = 40). Normotensive rats were used as controls (n = 12). In order to achieve this goal, animals were randomized selected and kept either under standard conventional housing conditions or received free access to a running wheel as a model of voluntary exercise during the night starting at the age of 6 weeks (pre-hypertensive state of SHR; Day 0). To characterize the acute effects of voluntary exercise on the renal transcriptome, a subgroup of rats was sacrificed already two days after the start of the trial (Day 2; n = 15). After six months, two groups of SHRs received the aldosterone blocker spironolactone (50 mg/kg/day via the feed) for another four months (n = 6 each); furthermore, the effects of performing physical activity have also been studied at the end of 6 months of training by keeping the animals under observation for an additional four months without running wheels (n = 6). After six and ten months, spot urine and plasma samples were collected from rats to monitor kidney function. An overview of the study protocol is given in Appendix A. Running distance, running time, and blood pressure were reported earlier for most of the rats used in this study [14,22]. Voluntary wheel running represents a type of aerobic exercise in which rats repeat short running boots of approximately 2–4 min.

At the end of the experimental protocol, rats were anesthetized by isoflurane inhalation. After cervical dislocation, kidneys were extracted, weighted, and immediately transferred to fluid nitrogen and stored at −80 °C or fixed for histological examination.

### 2.2. Blood Sampling

Blood samples (200 µL) were collected during the morning hours at the two indicated time points (month 6 and one week before rats were sacrificed) by punctuation of the tail vein. Plasma samples were generated by centrifugation in citrated tubes and subsequently stored at −80 °C until use. Spot urine was collected at the same time and immediately stored at −80 °C.

### 2.3. ELISA

Blood urea nitrogen (BUN) was analyzed with a DetectX Urea Nitrogen (BUN) Colometric Detection Kit (Arbor Assays, Ann Arbor, MI, USA) in plasma samples diluted 1:10 in sterile water. From plasma samples, renin concentration was analyzed by Rat renin-1 ELISA Kit (Sigma-Aldrich, St. Louis, MO, USA). In this case, plasma samples were diluted 1:10 with Sample Diluent C from the kit. From spot urine, Kim-1 was analyzed by Rat Kim-1 simpleStep ELISA kit (Abcam, Cambridge, UK); samples were diluted 1:5 with sample diluent NS. Microalbumin was quantified by Rat Microalbumin ELISA (Kamiya Biomedical Comp., Tukwila, WA, USA) in spot urine diluted 1:500 with diluent from the kit. Creatinine was analyzed in spot urine diluted 1:50 with sterile water and quantified by Creatinin PAP (Eberhardt Lehmann GmbH, Berlin, Germany). Analyses were performed by ELISA Reader infinite M200 (Tecan, Männedorf, Switzerland). Reads were evaluated by 4PL (myassay.com; accessed on 19 July 2019).

### 2.4. Histology

Kidney samples were fixed and pre-incubated with Tissue-Tek (Sakura, Alphen, The Netherlands) and sliced into 10 nm pieces. Slices were fixed in Bouin solution and subsequently stained in 0.1% (*w*/*v*) Sirius red solution (Sigma-Aldrich Chemie, Steinheim, Germany). Slices were washed by 0.01 N HCl and Aqua destillata for 5 min and dehydrated with ethanol. Finally, tissue slices were visualized under light microscope. Quantification was performed by Leica Confocal Software Lite Version 2.6.1 (Leica Microsystems CMS GmbH; Leica, Wetzlar, Germany). The mean of three preparations was used to quantify the extent of collagen labeling.

### 2.5. RT-PCR

Total RNA was isolated from kidneys using peqGOLD TriFast according to the manufacturer’s protocol. To remove genomic DNA contamination, isolated RNA samples were treated with 1 U DNase/µg RNA for 15 min at 37 °C. One microgram of RNA was used in a 10 µL reaction to synthesize cDNA using Superscript RNase H Reverse Transcriptase and oligo(dT) as primers. RT-PCR was performed as described before [14]. The sequences of the primers used in this study are indicated in Appendix A. Quantification was based on the ΔΔC_T_ method and performed as described before [30].

### 2.6. Statistcs

Data are expressed as means ± S.D. or presented as box plots with whiskers, where the 25, 50, and 75 percentiles are given by boxes, and the ranges are given by whiskers with individual data points within the range given as points. Exact *p* values are given, or *p* values below 0.05 are highlighted by *. *p* values were calculated by ANOVA with Student–Newman–Keuls post hoc analysis (Figures 1, 3, 4 and 6–8) or unpaired *t*-tests (Figures 3 and 5, Table 1). Effect sizes were analyzed by Cohen’s d. Group sizes were calculated on the basis of blood pressure alterations. Based on the variability of blood pressure in normotensive rats, we calculated the group sizes to detect differences in blood pressure of 20 mmHg between groups (two side effects, normal distribution). Calculations were based on the formula: 2 × (1.96 + 1.28)^2^ × (11/20)^2^. 2 (two side effect), 1.96 (97.5 quantile based on a *p*-value of 0.05), 1.28 (90% quantile based on a statistical power of 90%), 11 represents the standard deviation of blood pressure values of normotensive rats, 20 represents the calculated differences that should be achieved. Calculations end up with a number of 6.3 animals per group.

## 3. Results

### 3.1. Long-Term Adaptation of the Kidney to Voluntary Running Wheel Activity in SHRs

First, we sought differences in renal function between SHRs with voluntary running wheel activity and SHRs kept under sedentary conditions. For this reason, spot urine and blood plasma were collected from SHRs kept under sedentary conditions (SHR-S) and those with access to voluntary running wheel activity (SHR-R). Urinary secretion of Kim-1 (uKim-1) is a marker of tubular damage. In spot urine samples from young SHRs, we did not see an effect of two days of voluntary running wheel activity on uKim-1 (Figure 1A). However, after six months, uKim-1 was higher in SHRs performing voluntary running wheel exercise compared to sedentary control SHRs (Figure 1A; *p* = 0.025). Similarly, urinary secretion of albumin was not different between sedentary rats and rats performing voluntary running wheel activity after two days in young SHRs but increased after six months in the voluntary running wheel activity group (Figure 1B; *p* = 0.048). Plasma blood urea nitrogen (BUN) was not affected by voluntary running wheel activity in young and adult SHRs (Figure 1C).

A histological analysis of the kidney from sedentary SHRs and SHRs performing voluntary running wheel exercise did not show strong differences in glomerular structure but slightly thinner vessel walls in SHRs performing voluntary running wheel activity (Figure 2).

Compared to age-matched normotensive Wistar rats, ten-month-old SHRs had a slightly increased kidney-to-bodyweight ratio (effect size: 0.884 (95% CI: −0.085–1.834; *p* = 0.074)), strongly increased proteinuria (effect size: 2.229 (95% CI; 0.989–3.429; *p* = 0.000019)), and increased plasma BUN concentrations (effect size: 2.405; 95% CI: 1.191–3.581; *p* = 0.000065; Table 1). Voluntary running wheel activity did not affect these parameters in normotensive rats but reduced plasma BUN concentration in SHRs (effect size: 1.087; 95% CI: 0.078–2.071, *p* = 0.034; Table 1). uKIM-1 was increased in SHRs (effect size: 1.203; 95% CI: 0.124–2.251; *p* = 0.029) and normotensive rats (effect size: 0.920; 95% CI: −0.361–2.156; *p* = 0.163) by voluntary running wheel activity (Table 1).

These data indicate that voluntary running wheel activity is associated with moderate functional deficits as indicated by increased uKim-1 and microalbuminuria. Whether the expressions of molecular markers normally linked to renal dysfunction are also associated with moderate functional adaptations was investigated next.

Transcriptional adaptations to voluntary running wheel activity of the kidney were investigated in young SHRs after two days of voluntary running wheel activity. The mRNA expression of ODC1, PCSK9, HAVCR1, ARG2, LDLR, and REN increased with the onset of voluntary running wheel activity (Figure 3; *p* < 0.05). Interestingly, with the exception of ARG2 and PCSK9, all these inductions were normalized in adult SHRs with continuously voluntary running wheel activity (ODC1, HAVCR1, LDLR) or converted into a slight reduction (REN) (Figure 3).

Among the aforementioned molecular markers, renal mRNA expressions of *ODC1*, *HAVCR1*, and *REN* were elevated in normotensive sedentary rats compared to sedentary SHRs. Voluntary running wheel activity reduced renal mRNA expression of *REN* in normotensive rats (Figure 3). Alterations of renal renin expression were conformed at the level of renin plasma concentration. Plasma concentration of renin was higher in normotensive rats than in SHRs and reduced by voluntary running wheel activity in normotensive rats (Figure 4, *p* < 0.05).

### 3.2. Effect of Voluntary Running Wheel Activity on PCSK9 Expression in SHRs

The data on mRNA expression indicated a doubling of renal PCSK9 mRNA expression in SHRs but not in normotensive rats performing voluntary running wheel exercise. This leads to the question of whether this translates into protein expression. Western Blots indicate a similar expression pattern between normotensive Wistar rats and sedentary SHRs, but in SHRs performing voluntary running wheel activity, there was a strong increase in the protein expression of the 55 kDa subunit (Figure 5, *p* < 0.05). As this is the cleaved, secreted, and active subunit of PCSK9, the data suggest activation of PCSK9 in the kidney leading to a subsequent increase in enzymatic active PCSK9 as well.

### 3.3. Reversibility of Running-Induced Effects of SHRs

On the transcriptional level, the effect of voluntary running wheel activity in SHRs was most prominent on *PCSK9*, *ARG2*, and *REN*. To further validate the relationship between physical activity and the expression of these markers, a subgroup of SHRs was exposed to voluntary free running wheel activity for 6 months and subsequently kept under sedentary conditions for the last 4 months (ex runners; SHR-Ex). The data show that the effect of exercise on renal expression of *REN* and *ARG2* is strictly dependent on activity performance, whereas the increase in *PCSK9* remained high in SHR-Ex (Figure 6; *p* < 0.05). The effects of running wheel activity on plasma BUN concentration and uKim-1 were also reversed (Table 1). However, there was a trend to improvement of proteinuria in SHR-Ex (Table 1). Finally, we found a trend for induced *REN* expression in the kidney of Ex-Run SHRs (Figure 6).

The renal expression profile mimicked that of plasma renin levels in these rats, whereas the protein expression of arginase-2 (Figure 7) was not affected by increased mRNA expression suggesting an exercise-dependent increase in the turnover of arginase-2.

### 3.4. Effect of Aldosterone Blockade on Renal Adaptation and the Interaction with Voluntary Running Wheel Activity

As mentioned above, high physical activity can reduce the aldosterone concentration in the plasma. Therefore, we postulated that aldosterone blockade could mimic the effects of voluntary running wheel activity on renal function and molecular adaptation in SHRs. However, the effect of spironolactone on renal function significantly differed from that of high physical activity as performed by voluntary running wheel activity. The main effect of spironolactone was on proteinuria (effect size: 1.197; 95% CI: 0.135–2.228; *p* = 0.009, Table 1), whereas neither plasma BUN nor uKIM-1 was affected by spironolactone (Table 1). Moreover, uKim-1 remained increased by voluntary running wheel activity in the co-presence of aldosterone (effect size: 1.922; 95% CI: 0.488–3.296; *p* = 0.009; Table 1). On the molecular level, spironolactone reduced the renal expression of *PCSK9* and attenuated its increase in SHRs performing voluntary running wheel activity. However, spironolactone did not affect the expression of either *REN* or *ARG2* (Figure 8). In contrast to voluntary running wheel activity, mRNA expression of *PCSK9* (Figure 8; *p* < 0.05) and protein expression of PCSK9 (Figure 8) were reduced in the presence of spironolactone. Of note, the comparison between *ARG2* mRNA and protein expression suggested that voluntary running wheel activity increased the turnover of arginase-2. In the co-presence of spironolactone, this seems to be maintained. However, as spironolactone attenuated the exercise-dependent up-regulation of arginase-2 mRNA, the exercise-dependent increase in arginase-2 turnover resulted in lower levels of arginase-2 (Figure 8).

## 4. Discussion

The present study shows that high physical activity and aldosterone blockade act differently but are not additive in the kidney of female hypertensive rats. Blockade of aldosterone receptor activation by spironolactone markedly reduced the renal expression of PCSK9, and this was associated with a marked reduction in proteinuria, whereas the plasma concentration of BUN, a classical marker of glomerular filtration, was not affected. In line with this observation, PCSK9 shows a positive correlation with proteinuria but not with glomerular filtration in patients with chronic kidney disease [31,32]. Mechanistically, PCSK9 inhibits fatty acid β-oxidation-related factors, and this altered metabolism affects renal fibrosis [33].

In contrast, voluntary wheel running ameliorated plasma BUN, but we could not identify a molecular marker that was strongly associated with this improvement. Arginase-2 mRNA expression followed the pattern of BUN strictly. Most importantly, BUN and arginase-2 mRNA levels normalized four months after cessation of running wheel exercise. However, on the protein level, there was no correlation between arginase 2 mRNA and protein expression. An increase in arginase-2 mRNA expression caused by exercise without a concomitant increase in protein expression suggests that exercise increases the rate of protein degradation of arginase-2. In line with this assumption, we found that voluntary running wheel activity decreased arginase-2 protein levels in the presence of spironolactone, thereby under conditions that did not induce arginase-2 mRNA expression. This may indicate an aldosterone-dependent up-regulation of arginase-2 transcription in the kidney and an exercise-dependent increase in arginase-2 protein degradation that is aldosterone-independent. This assumption is in line with the assumption that mineralocorticoids mainly affect transcription. A similar type of regulation for arginase-2 in the kidney was described before in the context of mercuric chloride intoxication [34]. Arginase-2 protein degradation was induced without any effect on transcription or direct effect on enzyme activity. Therefore, increasing arginase-2 protein degradation seems to be a usual response to modify kidney function. In the paper cited above, overexpression of arginase-2 blocked mercuric chloride-dependent apoptosis. In our SHRs performing voluntary running wheel activity, mRNA expression of arginase-2 increased, and this was sufficient to stabilize arginase-2 protein levels but not to increase them. In the kidney, arginase-2 is predominantly expressed in the cortical and outer medullary proximal straight tubules [35]. In accordance with previous reports, renal arginase-2 expression in SHRs does not differ from that in normotensive Wistar rats [36]. High arginase-2 expression is associated with lower renal function and lower life expectancy [37,38]. This suggests that the downregulation of arginase-2 caused by voluntary wheel running is an adaptive mechanism. Nevertheless, the regulation of arginase-2 protein does not directly correlate with variations in any of the three markers of renal function (BUN, Alb/Cre, uKim-1). One may speculate that the decrease in arginase-2 protein expression by accelerated or increased degradation improves microcirculation, as suggested before [37], but on the other hand, it favors apoptosis, as also speculated before [34]. In summary, the findings on arginase-2 expression and voluntary running wheel activity are in accordance with other studies but are not associated with the observed effects of voluntary wheel running or aldosterone inhibition on renal function.

Voluntary running wheel activity was associated with increased uKim-1. Kim-1 is expressed in tubular epithelial cells, and it consists of an extracellular and a cytoplasmatic portion. The extracellular portion can be cleaved and enters the tubules if the kidney is injured [39]. Kim-1 can easily be detected in the urine, and its urinary concentration is closely related to kidney tissue damage. In our study, we found consistently higher levels of uKim-1 in all rats performing voluntary running wheel activity irrespectively of blood pressure and aldosterone inhibition. Therefore, the voluntary running wheel activity-dependent increase in uKim-1 was neither related to blood pressure nor to aldosterone. uKim-1 is a sensitive marker of renal stress that is independent of severe morphologic alterations [40]. In line with this, we did not see clear histological differences between kidneys from sedentary and exercise-performing SHRs. uKim-1 can be identified prior to changes in albuminuria or glomerular filtration and indicates the long-term prognosis [41,42,43]. As reported earlier, uKim-1 has no clear relationship to its renal expression and is lower expressed in SHRs compared to normotensive Wistar rats [39,44]. Collectively, the increased uKim-1 indicates renal stress caused by voluntary wheel running. Such stress due to exercise is also evident in microalbuminuria in humans and rodents [7,8,11,45,46]. It should be mentioned that increased uKim-1 is normalized by cessation of running wheel activity.

A major trigger of hypertension and hypertension-dependent disease is the overactivation of the renin-angiotensin-aldosterone system (RAAS). Renin was initially identified as a protease secreted from the kidney (reviewed in ref. [47]). However, it is now clear that renin can also be produced locally in the brain, affecting sympathetic activity and baroreflexes, and brain renin contributes to neurogenic hypertension (reviewed in ref. [48]). In the kidney, renin expression and secretion are regulated in a cAMP-dependent way [49]. As physical activity is associated with an activation of the sympathetic nervous system, it is likely to assume that this must affect the expression of renin in the kidney as well. Our study shows that renin is initially up-regulated in the kidney within two days when voluntary running wheel activity is started. However, constitutively high physical activity for months is associated with adaptations. One observation of such adaptations is the resting heart rate that is lower in rats performing high physical activity for months, indicating an increase in parasympathetic nervous system activity and, therefore, a reduction in sympathetic drive. Consequently, we observed a lower expression of renin in normotensive rats performing voluntary running wheel activity. Similarly to normotensive rats, the renin expression in the kidney in SHRs was also lower when performing voluntary running wheel activity for months. This effect was again reversible by cessation of voluntary running wheel activity. The renal expression of renin was also reflected in the renin plasma levels. This effect reduced renin levels in rats performing voluntary running wheel activity, which was blunted in rats receiving spironolactone. Nevertheless, renin mRNA and plasma levels were low in SHRs compared to normotensive rats. Given the importance of the RAAS in the development of hypertension in SHRs, this may be a surprise, but the data are in line with previous studies showing that SHRs compared to normotensive rats have less renin expression in the kidney [50,51,52] and release less renin into the circulation [53,54]. Thus, renin expression in the kidney may be of local importance rather than predominantly triggering the regulation of blood pressure. In line with this assumption, we found that the renin expression in SHRs strictly follows the changes in BUN, a marker of glomerular filtration. As normotensive Wistar rats have significantly higher levels of renin in the kidney and plasma, one would expect that they also have higher BUN levels. That was, however, not the case. An explanation for this finding is given in reports that components of the RAAS system, such as angiotensin-converting enzyme and angiotensin receptors, are increased in the kidney of SHRs compared to normotensive rats [55]. The kidney of SHRs may be more sensitive to RAS components than that of normotensive rats. Collectively our data suggest that a modification of the local renin expression by exercise is associated with improvements in BUN. Unexpectedly, this effect was blunted in the co-presence of spironolactone.

## 5. Conclusions

In conclusion, our study identified beneficial effects of aldosterone blockade and voluntary running wheel activity on renal function but no meaningful interaction between them. It was reported before that inhibition of the angiotensin-converting enzyme combined with physical activity has additive reno-protective effects in rodents [56]. However, in the cited study, the trigger of chronic kidney disease, adenine, caused morphological defects that were successfully ameliorated by co-treatment. In our study, morphological parameters were minimally altered, but functional parameters were significantly changed in SHRs versus normotensive rats. Among them, Alb/Cre was strongly ameliorated due to aldosterone blockade, and the increase in renal PCSK9 expression was also suppressed. The data are in accordance with the assumption that PCSK9 affects fatty acid metabolism in the kidney and thereby modifies the extracellular matrix [33]. In contrast, voluntary wheel running was characterized by an increase in uKim-1, indicating moderate stress on kidney function when physical activity was high. Our data also suggest that changes in sympathetic nervous activity observed in rats performing more physical activity have a direct influence on renal expression of renin, and this may affect local effects caused by renin release rather than being important for blood pressure control. In principle, lower renin expression in the kidney contributes to ameliorated BUN levels, but the fact that spironolactone blunted this influence suggests that it is at least in part dependent on aldosterone. Another interesting finding of our study is the effect of physical activity on arginase-2 protein degradation. We could not correlate this finding with meaningful functional parameters. However, high arginase-2 activity was repeatedly described to be associated with NO uncoupling affecting the microcirculation in the kidney [37,57]. Moreover, arginase-2 deficiency prolongs survival in rodents [38]. Therefore, arginase-2 protein degradation induced by physical activity should be considered a protective mechanism. In summary, our study clearly shows that a life-long activity, as it is represented by voluntary wheel running in this model, does not per se protect the kidney against hypertension-associated stress as indicated by increased plasma BUN and urinary KIM in our study. Moreover, the additive stress cannot be approved by additional aldosterone blockade except for the effect on urinary albumin. Future studies dealing with the interaction between physical activity and hypertension-dependent end-organ damage should include long-term analysis as an initial adaptive and protective effect may be lost by a continuation of exercise. Moreover, physical activity alone does not affect all regulatory pathways that are involved in hypertension, as it does not reduce blood pressure per se and does not attenuate all alterations. Future studies must focus on the unclear situation between plasma renin activity and renal expression of renin in hypertensive rats.

## Figures and Tables

**Figure 1 cells-11-03954-f001:**
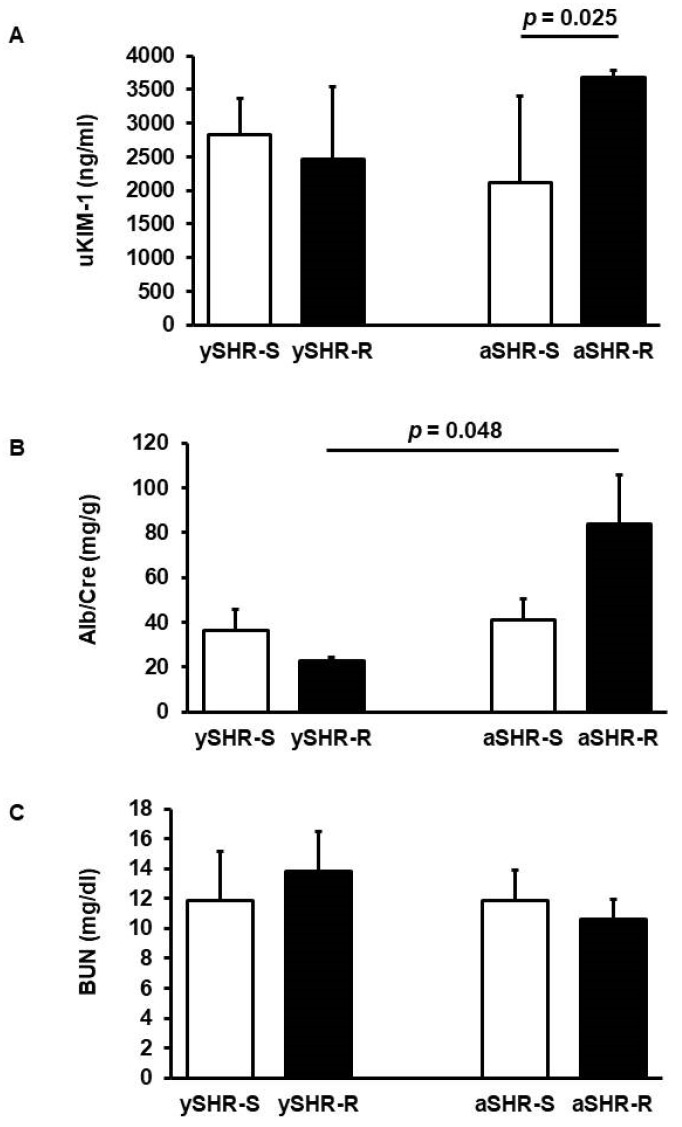
Effect of voluntary running wheel activity on (**A**) urinary Kim-1 secretion (uKIM-1), (**B**) microalbuminuria (Alb/Cre), and (**C**) plasma blood urea nitrogen (BUN). Data are given for young SHR (6 weeks, 2 days of voluntary running wheel activity) and adult SHR (7.5 months, 6 months voluntary running wheel activity). Rats were kept under standard housing conditions (sedentary, S) or had voluntary access to running wheels during the night (run, R). Data are means ± SD. ySHR-S (n = 5), ySHR-R (n = 10), aSHR-S (n = 6), aSHR-R (n = 6). Statistical analysis was performed by One-way ANOVA and subsequent Student–Newman–Keuls test.

**Figure 2 cells-11-03954-f002:**
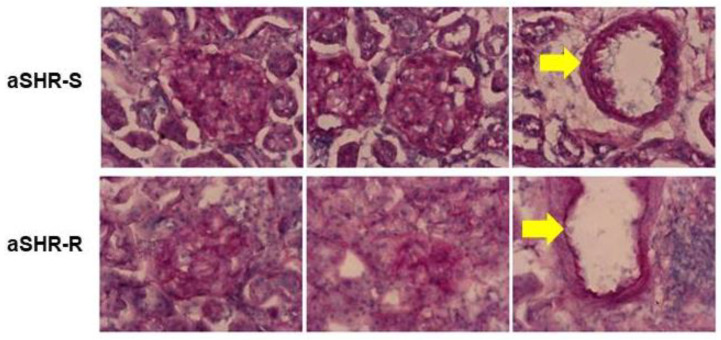
Representative periodic acid-Schiff reaction (PAS) staining of kidney slices from adult (7.5 months old) SHRs kept under standard housing conditions (S) or with access to running wheels (R). The major difference is the weaker staining of the vessel wall in the running group, as indicated by a yellow arrow.

**Figure 3 cells-11-03954-f003:**
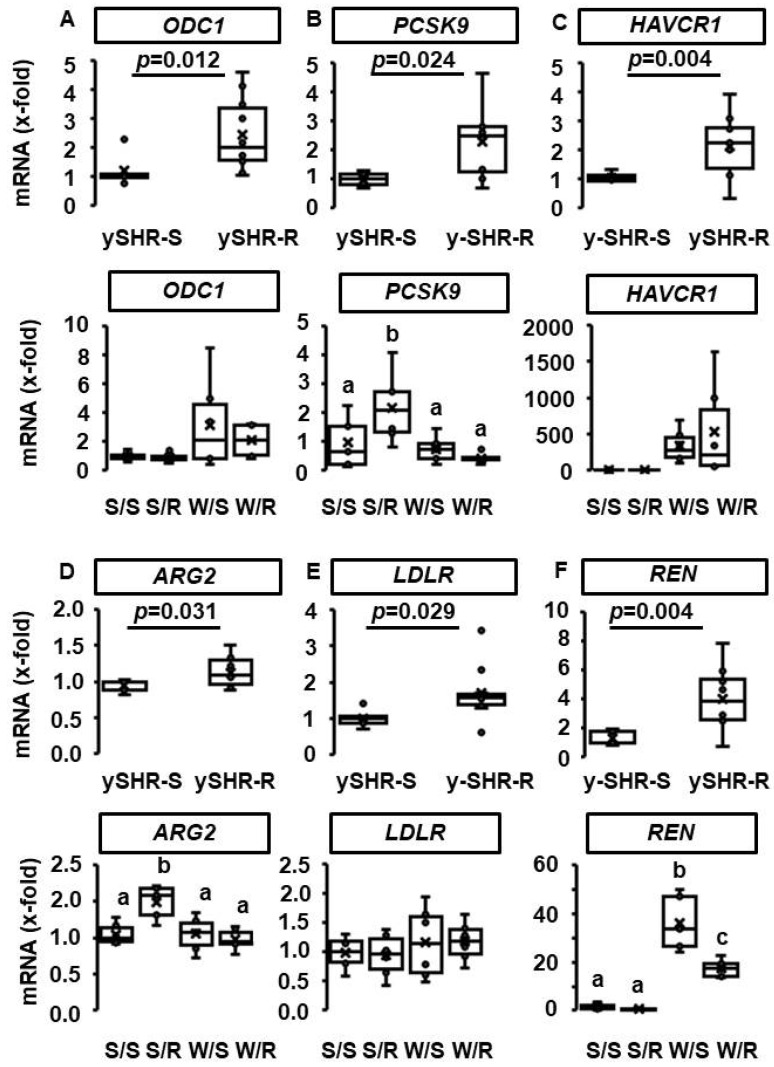
Transcriptional adaptation of the kidney to voluntary running wheel activity (R) compared to sedentary controls (S). Six mRNAs were identified with increased expression in young rats (6 weeks) after two days of voluntary running wheel activity in the running group: (**A**): ODC1 coding for ornithine decarboxylase; (**B**): PCSK9 coding for proprotein convertase subtilisin/kexin type 9; (**C**): HAVCR1, coding for kidney injury molecule-1; (**D**): ARG2 coding for arginase-2; (**E**): LDLR coding for, low-density lipoprotein receptor; (**F**): REN coding for renin). Among them, only PCSK9 and ARG2 remained increased after ten months of voluntary running wheel activity (bottom line). Data from adult SHRs were compared to normotensive Wistar rats (W). Data are expressed as box and whiskers, where the 25, 50, and 75 percentiles are given by boxes, and the ranges are given by whiskers with individual data points within the range given as points. Exact *p* values are given for young rats (*t*-Test) and group differences for adult rats (one-way ANOVA and Student–Newman–Keuls test). Small letter indicate groups within *p* > 0.05. For n numbers of the different groups, see Appendix A.

**Figure 4 cells-11-03954-f004:**
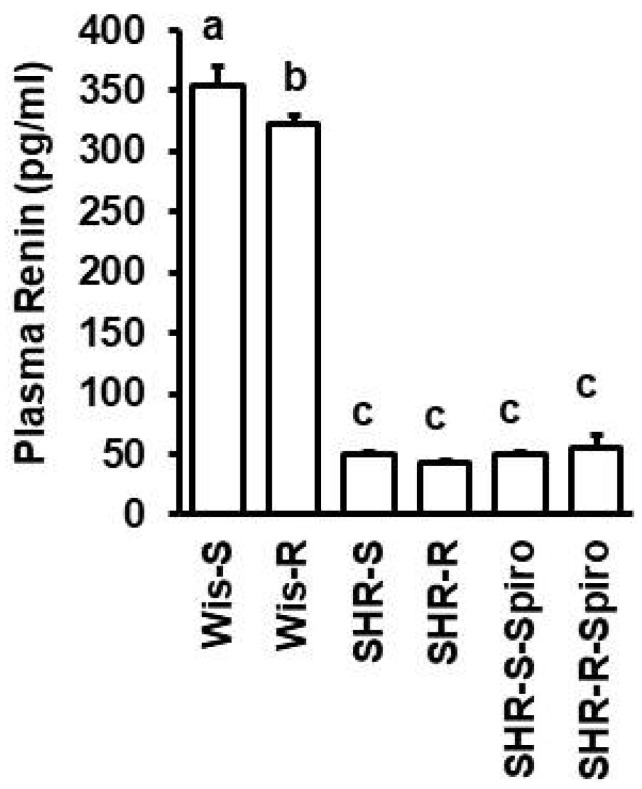
Plasma renin concentrations in normotensive Wistar rats (Wis) and SHRs under sedentary control conditions (S), with free access to running wheels (R), and with the administration of spironolactone (Spiro). Plasma samples were collected after ten months. Data are means ± S.D. with one-way ANOVA and Student–Newman–Keuls post hoc test. Group differences are indicated by letters. For n numbers of the different groups, see Appendix A.

**Figure 5 cells-11-03954-f005:**
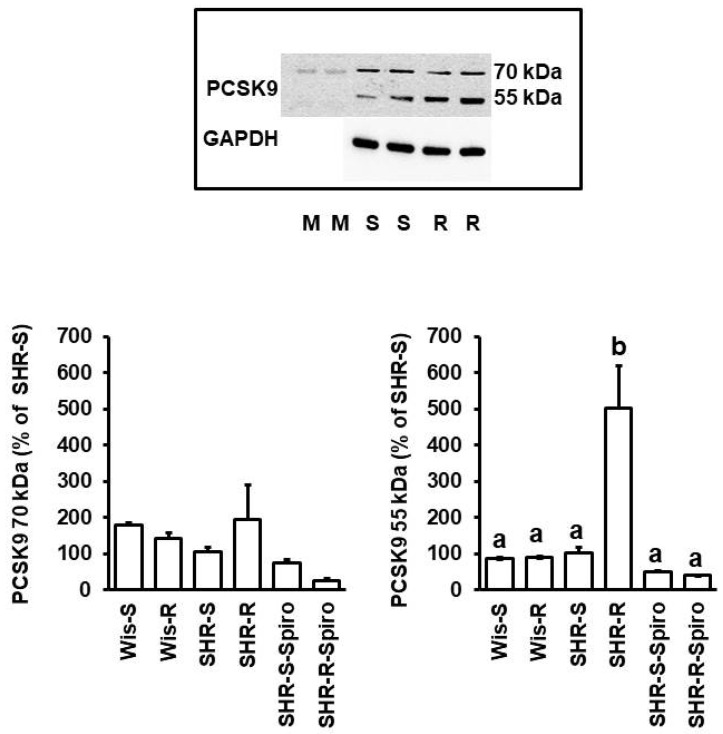
PCSK9 protein expression in rats shown in Figure 4. **Top**: Original Western blot showing the two bands of PCSK9. M: marker; S: sedentary SHRs; R: SHRs with voluntary running wheel exercise. **Bottom**: Quantification of the expression PCSK9. Data are means ± S.D. with one-way ANOVA and Student–Newman–Keuls post hoc test. Group differences are indicated by letters. For n numbers of the different groups, see Appendix A.

**Figure 6 cells-11-03954-f006:**
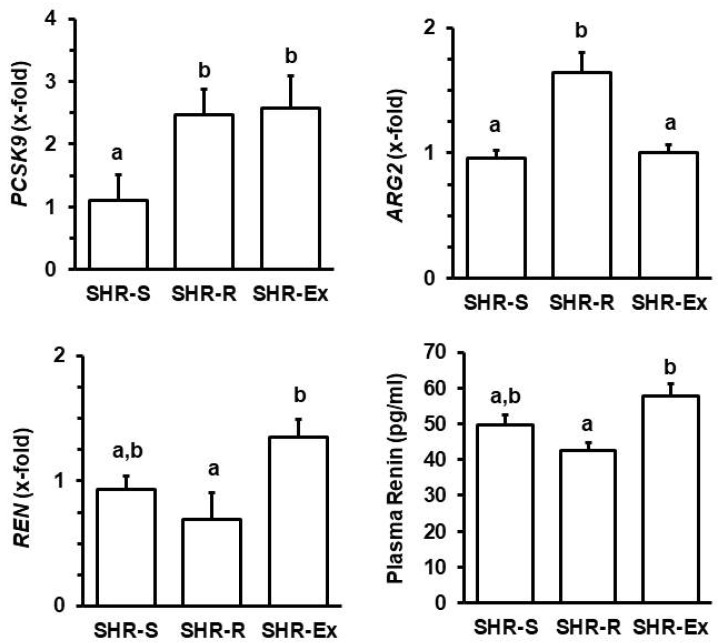
Effect of running cessation on mRNA expression of *PCSK9*, *ARG2*, and *REN* and plasma renin concentration in SHRs kept under standard conditions (S), with free access to running wheels (R) or voluntary running wheel activity for 6 months and subsequently four months of sedentary housing (Ex). Data are means ± S.D. with one-way ANOVA and Student–Newman–Keuls post hoc test. Group differences are indicated by letters. For n numbers of the different groups, see Appendix A.

**Figure 7 cells-11-03954-f007:**
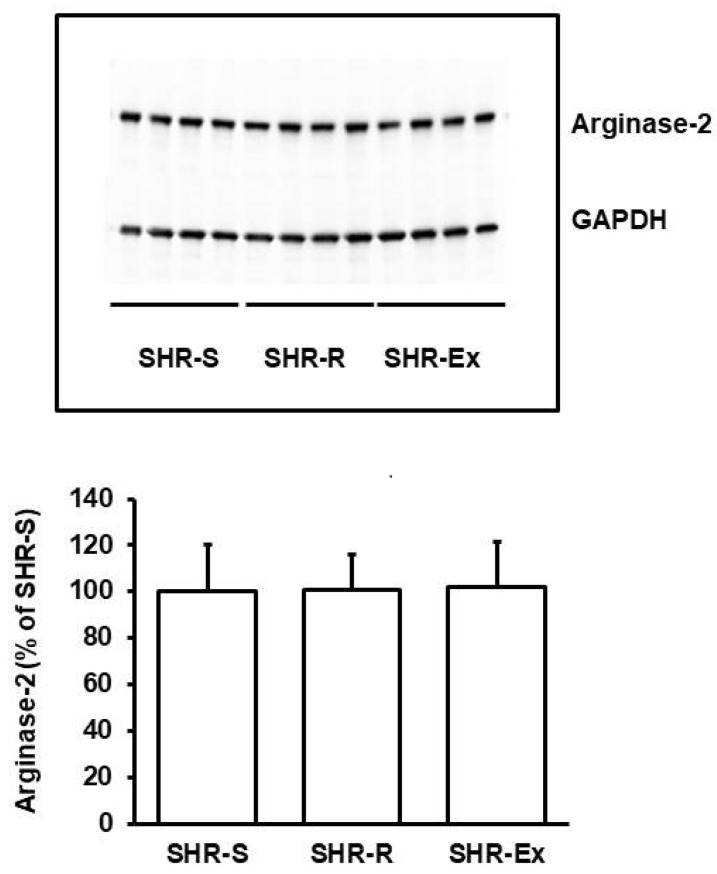
Western blot indicating the protein expression of arginase-2 in rats, as in Figure 6. **Top**: Representative immunoblot; **Bottom**: Quantification of the samples. Data are means ± S.D. Statistical analysis was performed by one-way ANOVA (*p* > 0.05).

**Figure 8 cells-11-03954-f008:**
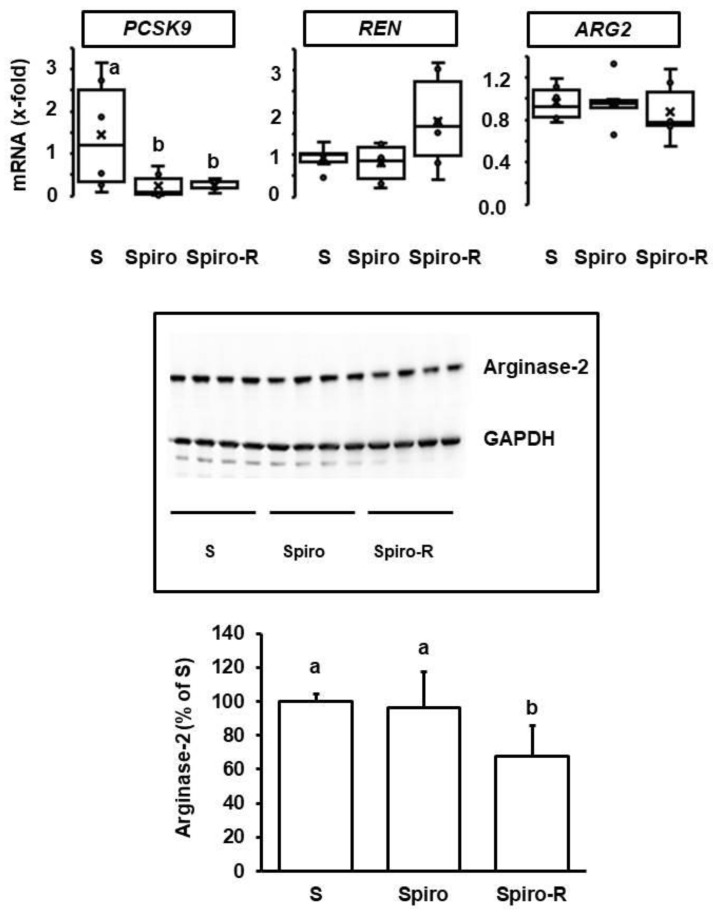
Effect of spironolactone on *PCSK9*, *REN,* and *ARG2* expression in SHRs kept under standard conditions (S) or free access to running wheels. Top: mRNA expression. Data are expressed as box and whiskers, where the 25, 50, and 75 percentiles are given by boxes, and the ranges are given by whiskers with individual data points within the range given as points. Statistical analysis was performed by one-way ANOVA (*p* > 0.05). Bottom: Representative immunoblot and quantification of the blot. Data are means ± S.D. with subsequent statistical analysis by one-way ANOVA and Student–Newman–Keuls post hoc analysis. Group differences are indicated by letters.

**Table 1 cells-11-03954-t001:** Kidney wights and function after 10 months.

	KW/BW (mg/g)	Alb/Cre (mg/g)	BUN (mg/dL)	uKIM (ng/mL)
WIS-S	3.15 ± 0.09	18.97 ± 8.27	12.50 ± 1.45	1745 ± 405
WIS-R	3.13 ± 0.37	17.66 ± 8.58	13.87 ± 1.12	2761 ± 1301
SHR-S	3.33 ± 0.23	231.29 ± 120.55 #	19.08 ± 5.31 #,$	2092 ± 1179
SHR-R	3.31 ± 0.11	207.52 ± 50.80	15.91 ± 2.36	3567 ± 1108 *
SHR-Ex	3.33 ± 0.26	141.45 ± 44.09	18.01 ± 2.06	2183 ± 1359
SHR-S-Spiro	3.40 ± 0.26	110.43 ± 57.22	18.70 ± 1.63	1568 ± 949
SHR-R-Spiro	3.12 ± 0.09	86.37 ± 46.91	20.55 ± 2.71	3757 ± 1301 *,$

Means ± S.D. from n = 18 (SHR-S) or n = 6 (all other groups). *, *p* < 0.05 run vs. sedentary; #, *p* < 0.05 vs. WIS-S; $, *p* < 0.05 vs. SHR-S.

## Data Availability

Data are stored on the central computer of the Physiologisches Institute of the JLU Gießen, Germany. All data generated in the context of this project are reported. Individual data requests can be sent to the corresponding author.

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
