# Peer review of "Effect of Metabolic Adaptation by Voluntary Running Wheel Activity and Aldosterone Inhibition on Renal Function in Female Spontaneously Hypertensive Rats"

_cells, 2022, doi:10.3390/cells11243954_

Round 1

Reviewer 1 Report

Starting with the notion that exercise can affect renal function in mice, rats, and humans with healthy or disease status, the authors have analyzed interaction between hypertension, aldosterone and physical activity on kidneys by using spontaneously hypertensive rats with the model with or without aldosterone blockade. Independent but no additive effects of aldosterone blockade and physical activity on renal function and on molecules potentially affecting renal lipid metabolism have been observed. It seems that in presented model of essential hypertension kidney function does not benefit from a combination of high physical activity and aldosterone blockade.

The authors have used several well chosen and appropriately applied methods to elucidate the proposed assumptions. All required ethical prerequisites have been respected. The obtained data were adequately analyzed and comprehensively discussed in the light of previous results obtained by other authors. This study will certainly increase the current knowledge of investigated problem and increase our expertise in the sense of future clinical implications.

Very well prepared and executed study.

Author Response

We thank the reviewer for his/her time spent to review the manuscript and for the comments.

Reviewer 2 Report

Dear ,

Manuscript Number: Cells- 2037766

Title Manuscript: Effect of metabolic adaptation by voluntary running wheel activity and aldosterone inhibition on renal function in female spontaneously hypertensive rats

This experimental study is an important topic in the field of exercise physiology with animal models on renal function, and the text of this manuscript is well written but at the moment MAJOR REVISIONS are necessary in order to make it suitable for a final decision for “Cells”.

POINTs of STRENGTH:

1) Assessment of the effects of long-term high physical activity (10 months of voluntary running wheel activity) on renal function and metabolic adaptations in female animal models (an experimental study);

2) Arguments in the discussion section;

POINTs of WEAKNESS (and/or should be revised to improve the manuscript):

Abstract:

3) The purpose of this experimental study is not specified in the objectives section; please specify clearly;

4) The number and classification of rates in different groups are not specified in the methods section of the Abstract; please clarify;

5) Please provide the mean age and weight of male rats as well as gender in the methods section of the Abstract;

6) Please specify the study protocol and/or duration of voluntary running wheel activity and its protocol as well as “aldosterone blocker spironolactone” intervention in the methods section of the Abstract;

Keywords

7) Please add the keywords of “Voluntary wheel running” and “spironolactone”

Introduction:

8) The hypothesis and purpose of this study can be stated in more detail;

2. Materials and methods

2.1. Animal Model

9) Please specify the total rats and number of rats in each group in the animal model section;

10) Please specify the type of exercise (aerobic and/or endurance) for Voluntary wheel running activity;

2.2. Blood Sampling

11) Please specify the time (morning or other time, and hour) and amount of blood sampling;

2.6. Statistcs (statistcs or statistics; please correct)

12) Did authors use a statistical software to calculate the sample size? IF YES, please explain and add its name and valid reference in the statistical analysis section.

3. Results

13) Please specify the significance level for the results in the text of the manuscript;

4. Discussion

14) High physical activity or voluntary wheel running? Please clarify in the discussion section;

5. Conclusions

15) What are the conclusions and implications for future research?;

16) What does this experimental study add to the literature?;

References

17) References section is not always in accordance with the authors' guidelines. In particular, please check No. 13, 24, and 36 for validation.

Best Regards

7 November 2022

Author Response

Abstract:

3) The purpose of this experimental study is not specified in the objectives section; please specify clearly;

Response: This study was designed to evaluate whether physical activity and pharmacological treatment by aldosterone have additive effects on renal protection in hypertensive rats.

4) The number and classification of rates in different groups are not specified in the methods section of the Abstract; please clarify;

Response: The following groups were studied: Young and pre-hypertensive SHR (n=5 sedentary; n=10 running wheels, mean body weight 129 g), 10 months old Wistar rats (n=6 sedentary; n=6 running wheels, mean body weight 263 g), 10 months old SHRs (n=18 sedentary, mean body weight 224 g; n=6 running wheels, mean body weight 272 g; n=6 aldosterone, mean body weight 219 g; n=6 aldosterone and running wheels, mean body weight 265 g). Another group of SHRs had free access to running wheels for 6 months and kept sedentary for the last 4 months (n=6, mean body weight 240 g).

5) Please provide the mean age and weight of male rats as well as gender in the methods section of the Abstract;

Response: see Response to Q4!

6) Please specify the study protocol and/or duration of voluntary running wheel activity and its protocol as well as “aldosterone blocker spironolactone” intervention in the methods section of the Abstract;

Response: Aldosterone was given for the last 4 months. SHRs from the running groups had free access to running wheels beginning at an age of 6 weeks.

Keywords

7) Please add the keywords of “Voluntary wheel running” and “spironolactone”

Response: Done

Introduction:

8) The hypothesis and purpose of this study can be stated in more detail;

Response: … we tested the hypothesis that the level of physical activity directly affects the molecular adaptations of the kidney to hypertension and that these effects are independent of aldosterone. 

  1. Materials and methods

2.1. Animal Model

9) Please specify the total rats and number of rats in each group in the animal model section;

Response: Done

10) Please specify the type of exercise (aerobic and/or endurance) for Voluntary wheel running activity;

Response: Voluntary wheel running represents a type of aerobic exercise in which rats repeat short running boots of approximately 2-4 minutes.

2.2. Blood Sampling

11) Please specify the time (morning or other time, and hour) and amount of blood sampling;

Response: Done

2.6. Statistcs (statistcs or statistics; please correct)

12) Did authors use a statistical software to calculate the sample size? IF YES, please explain and add its name and valid reference in the statistical analysis section.

Response: Group sizes were calculated on the basis blood pressure alterations. Based on the variability of blood pressure in normotensive rats we calculated the group sizes to detect differences in blood pressure of 20 mmHg between groups (two side effect, normal distribution). Calculations were based on the formula: 2 x (1.96 + 1.28)2 x (11 / 20)2. 2 (two side effect), 1.96 (97.5 quantile based on a p-value of 0.05), 1.28 (90 % quantile based on a statistical power of 90%), 11 represents the standard deviation of blood pressure values of normotensive rats, 20 represents the calculated differences that should be achieved. Calculation end-up with a number of 6.3 animals per group.

  1. Results

13) Please specify the significance level for the results in the text of the manuscript;

Response: Done

  1. Discussion

14) High physical activity or voluntary wheel running? Please clarify in the discussion section;

Response: High physical activity replaced by voluntary wheel running in the Discussion part.

  1. Conclusions

15) What are the conclusions and implications for future research?;

16) What does this experimental study add to the literature?;

Response: We replaced our final summary. In summary, our study clearly shows that a life-long activity as it is represented by voluntary wheel running in this model does not per se protect the kidney against hypertension-associated stress as indicated by increased plasma BUN and urinary KIM in our study. Moreover, the additive stress cannot be improved by additional aldosterone blockade except an effect on urinary albumin. Future studies dealing with the interaction between physical activity and hypertension-dependent end-organ damage should include long-term analysis as an initial adaptive and protective effect may be lost by continuation of exercise. Moreover, physical activity alone does not affect all regulatory pathways that are involved in hypertension as it does not reduce blood pressure per se and does not attenuate all alterations. Future studies must focus on the unclear situation between plasma renin activity and renal expression of renin in hypertensive rats.

References

17) References section is not always in accordance with the authors' guidelines. In particular, please check No. 13, 24, and 36 for validation.

Response: Done

Round 2

Reviewer 2 Report

Dear,

Manuscript Number: cells-2037766

Title Manuscript: Effect of metabolic adaptation by voluntary running wheel activity and aldosterone inhibition on renal function in female spontaneously hypertensive rats

I am very grateful to the authors for their efforts.

In general, this manuscript has found suitable content after correcting major revisions, and the modified revisions are accepted.

Best Regards

30 November 2022